# Degradation Products of Complex Arabinoxylans by *Bacteroides intestinalis* Enhance the Host Immune Response

**DOI:** 10.3390/microorganisms9061126

**Published:** 2021-05-22

**Authors:** Taro Yasuma, Masaaki Toda, Ahmed M. Abdel-Hamid, Corina D’Alessandro-Gabazza, Tetsu Kobayashi, Kota Nishihama, Valeria Fridman D’Alessandro, Gabriel V. Pereira, Roderick I. Mackie, Esteban C. Gabazza, Isaac Cann

**Affiliations:** 1Department of Immunology, Mie University, Tsu City 514, Mie, Japan; t-yasuma0630@clin.medic.mie-u.ac.jp (T.Y.); t-masa@doc.medic.mie-u.ac.jp (M.T.); dalessac@clin.medic.mie-u.ac.jp (C.D.-G.); kn2480@gmail.com (K.N.); immunol@doc.medic.mie-u.ac.jp (V.F.D.); 2Carl R. Woese Institute for Genomic Biology (Microbiome Metabolic Engineering Theme), University of Illinois at Urbana-Champaign, Urbana, IL 61801, USA; ahetta@illinois.edu (A.M.A.-H.); gpereira@umich.edu (G.V.P.); r-mackie@illinois.edu (R.I.M.); icann@illinois.edu (I.C.); 3Department of Botany and Microbiology, Faculty of Science, Minia University, El-Minia 61519, Egypt; 4Department of Pulmonary and Critical Care Medicine, Mie University, Tsu City 514, Mie, Japan; kobayashitetsu@hotmail.com; 5Department of Animal Science, University of Illinois at Urbana-Champaign, Urbana, IL 61801, USA; 6Energy Biosciences Institute, University of Illinois at Urbana-Champaign, Urbana, IL 61801, USA; 7School of Molecular and Cellular Biology, University of Illinois at Urbana-Champaign, Urbana, IL 61874, USA; 8Department of Microbiology, University of Illinois at Urbana-Champaign, Urbana, IL 61801, USA

**Keywords:** Colonic *Bacteroidetes*, microbiome, dietary fiber, arabinoxylans, ferulic acid, health benefits

## Abstract

*Bacteroides* spp. of the human colonic microbiome degrade complex arabinoxylans from dietary fiber and release ferulic acid. Several studies have demonstrated the beneficial effects of ferulic acid. Here, we hypothesized that ferulic acid or the ferulic acid-rich culture supernatant of *Bacteroides intestinalis*, cultured in the presence of complex arabinoxylans, enhances the immune response. Ferulic acid and the culture supernatant of bacteria cultured in the presence of insoluble arabinoxylans significantly decreased the expression of tumor necrosis factor-α and increased the expression of interleukin-10 and transforming growth factor β1 from activated dendritic cells compared to controls. The number of granulocytes in mesenteric lymph nodes, the number of spleen monocytes/granulocytes, and interleukin-2 and interleukin-12 plasma levels were significantly increased in mice treated with ferulic acid or the culture supernatant of bacteria cultured with insoluble arabinoxylans. Ferulic acid or the culture supernatant of bacteria cultured with insoluble arabinoxylans increased the expression of interleukin-12, interferon-α, and interferon-β in intestinal epithelial cell lines. This study shows that ferulic acid or the ferulic acid-rich culture supernatant of the colonic bacterium *Bacteroides intestinalis*, cultured with insoluble arabinoxylans, exerts anti-inflammatory activity in dendritic cells under inflammatory conditions and enhances the Th1-type immune response under physiological conditions in mice.

## 1. Introduction

There is accumulating evidence showing the beneficial health effects of dietary fiber [1,2]. Whole grains (wheat, rye, oat, barley), vegetables, and fruits are the main sources of dietary fiber [3]. High consumption of whole grains, vegetables, and fruits reduces the risk of cardiovascular diseases, cancer, gastrointestinal disorders, and diabetes mellitus [1,2]. Furthermore, a meta-analysis of 367,442 subjects followed up for 14 years showed that whole grains consumption significantly decreases the risk of death from chronic diseases, including cardiovascular diseases, diabetes, respiratory diseases, and infections [4]. Arabinoxylan is the hemicellulose that accounts for approximately 70% of dietary fiber’s soluble and insoluble content [5]. Arabinoxylans form the structure of cell walls of plants, including cereal grains [6]. The insoluble form of arabinoxylans contains large amounts of phenolic compounds, including ferulic acid [7,8]. 

The rich content of ferulic acid has been attributed to the health benefits of arabinoxylans [9], and aside from its antioxidant function, ferulic acid has also been reported to exert modulatory activity on the inflammatory response [9]. Furthermore, recent studies have suggested that ferulic acid may also enhance the host defense mechanism by stimulating the immune system and physiological homeostasis [10]. In agreement with these observations, the administration of ferulic acid increased the number of circulating leukocytes, lysosome activity, and serum levels of antioxidants [10]. 

The release of ferulic acid from dietary fiber requires the degradation of complex arabinoxylans [7,11]. However, as humans lack the enzymes capable of degrading arabinoxylans, the degradation of this complex polysaccharide depends on enzymes released by members of the gut microbiome [12,13]. Importantly, previous studies have shown that enzymes secreted by some bacteria belonging to phyla *Bacteroidetes* and *Firmicutes* can digest arabinoxylans in the human lower gastrointestinal tract [13,14,15,16]. In addition, we recently demonstrated that the degradation of complex arabinoxylans by *Bacteroides* spp. releases large amounts of ferulic acid [7]. Based on these observations, we hypothesized that the degradation products of complex arabinoxylans by *Bacteroides intestinalis*, a member of the human colonic *Bacteriodetes*, would enhance the host immune response. To interrogate this hypothesis, we used both in vitro and in vivo approaches to determine the impact of bacterially released ferulic acid, from complex arabinoxylans, on inflammatory and physiological conditions of mice.

## 2. Materials and Methods

### 2.1. Animals

We used female C57BL/6 mice of 8-weeks of age and 20 g in weight from Japan SLC (Hamamatsu, Japan). The animals were maintained in a pathogen-free environment under a constant 12:12-h light–dark cycle in a temperature- and humidity-controlled room and given water and standard mouse food ad libitum. 

### 2.2. Ethical Statement

The Mie University’s Committee on Animal Investigation approved the experimental protocol (Approval No 28–21; 5 May 2019). To perform the experimental animal procedures, we followed the institutional guidelines of Mie University and the internationally approved principles of laboratory animal care published by the National Institute of Health (https://olaw.nih.gov/) (accessed on 2 May 2018). The research followed the ARRIVE Guidelines for animal investigation. We performed measurements blindly of the treatment groups.

### 2.3. Bacterial Strains

*Bacteroides intestinalis* 341 (DSM 17393) was a kind gift from Jeffrey I. Gordon (Washington University in St. Louis, MO, USA). This bacterial strain was originally obtained from the DSMZ (German Collection of Microorganisms and Cell Cultures, Braunschweig). 

### 2.4. Arabinoxylan Substrates 

Soluble wheat arabinoxylan with medium viscosity, insoluble wheat arabinoxylan, xylose, and arabinose were obtained from Megazyme (Bray, Ireland). Preparation of the de-starched wheat bran (DWB) was done as described in our earlier report [7]. Briefly, the DWB was prepared by milling 60 g of wheat bran (WB) to pass through a sieve of diameter < 0.5 mm, followed by suspension in 600 mL of phosphate buffer (80 mM, pH 6.2). We then added 4.5 mL of the starch-degrading thermostable alpha-amylase (Termamyl 120 L, EC 3.2.1.1, from *Bacillus licheniformis*, 20,000–60,000 U/mL) to the mixture, followed by incubation at 92 °C for 20 min. The insoluble fraction, i.e., the DWB, was obtained by centrifugation at 4000× *g* rpm for 10 min, followed by washing twice with water, twice with acetone, and finally drying at 40 °C. 

### 2.5. Bacterial Growth Conditions 

*Bacteroides intestinalis* was cultured in an anaerobic chamber with a gas mixture of 85% N_2_, 10% CO_2_, and 5% H_2_. The bacterium was cultured from a single colony inoculated into a medium supplemented with a brain–heart infusion (BHIS) at 37 g/L, hemin chloride (1.9 µM), cysteine hydrochloride at 4 mM, and sodium bicarbonate at 9.5 mM. For the growth experiment, 100 μL of cells cultured overnight in the BHIS medium were inoculated into a defined medium containing a specified carbon source (soluble wheat arabinoxylan or insoluble wheat arabinoxylan or xylose/arabinose or wheat bran or de-starched wheat bran) and passed through three dilutions to remove any residues derived from the growth in the BHIS medium. The medium contained sodium chloride (15 mM), dipotassium hydrogen phosphate (5 mM), monopotassium hydrogen phosphate (5 mM), sodium bicarbonate (9.5 mM), cysteine hydrochloride (4 mM), magnesium (II), chloride heptahydrate (0.1 mM), calcium (II) chloride dihydrate (54 µM), iron (II) sulfate heptahydrate (1.4 µM), hemin chloride (1.9 µM), vitamin K3 (5.8 µM), vitamin B12 (7.3 nM), and a specified carbon source (5 mg/mL).

### 2.6. Cell Culture

Reagents for the cell culture were purchased from Nacalai Tesque Inc. (Kyoto, Japan). Trans-ferulic acid was obtained from Sigma-Aldrich Co. (Tokyo, Japan). The human epithelial colorectal adenocarcinoma cell line Caco-2 and the non-transformed and non-immortalized human fetal small intestine cell line HIEC-6 were obtained from ATCC (Manassas, VA, USA). They were propagated and maintained in DMEM supplemented with 20% (*v*/*v*) heat-inactivated fetal bovine serum (FBS), 0.03% (*w*/*v*) L-glutamine, 100 IU/mL penicillin, and 100 μg/mL streptomycin (penicillin–streptomycin, Gibco) in a humidified, 5% CO_2_ atmosphere at 37 °C, and the medium was refreshed every three days. All experiments were conducted on days 10–12 post-seeding when cells reached confluence and became differentiated.

### 2.7. Treatment of Intestinal Caco-2 and HIEC-6 Cells

Caco-2 cell monolayers grown in 12-well plates (Corning Incorporated, Kennebunk, ME, USA) were serum-starved overnight, incubated in the presence of (0,1, 10, 100 µM) trans-ferulic acid or 1/2 volume of culture supernatants from *Bacteroides intestinalis*, and after 1 h, *Escherichia coli* lipopolysaccharide (LPS) was added at a final concentration of 10 µg/mL. Culture media were collected 24 h after ferulic acid administration to determine IL-12p70, interferon (IFN)α, and IFNβ by commercially available enzyme-linked immune sorbent assay kits (R&D Systems, Minneapolis, MN). Total RNA from Caco-2 cells at 24 h was isolated using Sepasol-RNA I Super G reagent (Nacalai). The RNA concentration and purity were determined by UV absorption at 260:280 nm using an Ultrospec 1100 pro UV/Vis spectrophotometer (Amersham Biosciences, NJ). RNA was reverse-transcribed into cDNA using a ReverTra Ace qPCR RT kit (TOYOBO, Osaka, Japan), according to the manufacturer’s protocol, and then the DNA was amplified by PCR using Quick Taq HS DyeMix (TOYOBO). The sequences of the primers and the number of PCR cycles are listed in Table 1. The PCR products were separated on a 2% agarose gel containing 0.01% ethidium bromide. The amount of mRNA was normalized against the glyceraldehyde 3-phosphate dehydrogenase mRNA.

### 2.8. Isolation and Treatment of Dendritic Cells 

After removing tibias and femurs from C57BL/6 WT mice, they were flushed with RPMI-1640 medium containing 10% heat-inactivated FBS. The debris was removed by passing through cotton wool, and the bone marrow-derived dendritic cells (BMDCs) were cultured for seven days at 37 °C in a humidified atmosphere containing 5% CO_2_ using RPMI-1640 medium. The culture medium was supplemented with 10% heat-inactivated FBS (BioWhittaker, Inc., Walkersville, MD, USA), 2 mM L-glutamine, 100 U/mL penicillin, 100 µg/mL streptomycin, 50 µM µ-mercaptoethanol (Sigma-Aldrich Co, St. Louis, MO, USA), and 200 ng/mL Flt3L (PeproTech, Inc., Rocky Hill, NJ, USA). CD11c+ BMDCs were then harvested using mouse CD11c microbeads (Miltenyi Biotec GmbH). BMDCs were then cultured in the presence or absence of 100 ng/mL of *Escherichia coli* LPS (Sigma-Aldrich Co) and stimulated with ferulic acid (FA); control medium containing the simple sugar components of arabinoxylan, i.e., arabinose/xylose (Ara/xylose); simple or soluble wheat arabinoxylan (sWAX) or insoluble wheat arabinoxylan (InWAX); or the culture supernatant from *Bacteroides intestinalis* (*B.i.*) cultured on Ara/xylose (i.e., *B.i.* + Ara/xylose), sWAX (i.e., *B.i.* + sWAX), or InWAX (i.e., *B.i.* + InWAX). Cell supernatants were then collected to measure tumor necrosis factor α (TNFα; BD Biosciences, San Jose, CA, USA), transforming growth factor β1 (TGFβ1; R&D Systems, Minneapolis, MN, USA), and interleukin-10 (IL-10; BD Biosciences, San Jose, CA, USA), using commercially available enzyme immunoassay kits.

### 2.9. Isolation and Analysis of Mesenteric Lymph Node Cells and Spleen Cells 

A volume of 0.2 mL of ferulic acid, culture supernatant from *Bacteroides intestinalis* grown in wheat bran (WB) or de-starched wheat bran (DWB), control medium with wheat bran or de-starched wheat bran, or distilled water was orally administered by gavage to mice (n = 3, each treatment group) every day for seven consecutive days before euthanasia of mice by anesthetic overdose. Mesenteric lymph nodes and spleens were removed, immediately incised, minced with scissors into 2–3 mm pieces, and incubated for 30 min at 37 °C in RPMI medium containing 0.5 mg/mL collagenase (Sigma-Aldrich) and 10 µg/mL DNAase (Sigma-Aldrich). Single-cell suspensions were then obtained by grinding and filtering the tissues through a 70 µm diameter nylon mesh (BD Bioscience, San Jose, CA). Mesenteric lymph node and spleen cells were analyzed by flow cytometry (FACScan, BD Biosciences, Oxford, UK).

### 2.10. Blood Collection 

Peripheral blood was collected by cardiac puncture under deep anesthesia and placed into tubes containing sodium heparin (14 U/mL), and plasma was separated after centrifugation and stored at −80 °C until analysis. 

### 2.11. Flow Cytometry Analysis 

The percentage of spleen cells was evaluated by flow cytometry (BD Biosciences, Oxford, UK) using the following antibodies: fluorescein isothiocyanate (FITC)-labeled anti-mouse Ly-6G/Ly-6C (Gr-1; clone RB6-8C5) rat IgG2bκ, phycoerythrin (PE)-labeled anti-mouse F4/80 (clone CIA3-1) rat IgG2bκ, PE/Cy5-labeled anti-mouse CD11c (clone N418) hamster IgG, FITC-labeled anti-mouse CD3ε (clone 145-2C11) hamster IgG, PE/Cy5-labeled anti-mouse CD45R/B220 (clone RA3-6B2) rat IgG2aκ, FITC-labeled anti-mouse CD25 (clone PC61) rat IgG1λ, PE-labeled anti-mouse CD8a (clone 53-6.7) rat IgG2aκ, and PE/Cy5-labeled anti-mouse CD4 (clone GK1.5) rat IgG2bκ, which were from BioLegend, Inc. (San Diego, CA). PE-labeled anti-mouse NK1.1 (clone PK136) mouse IgG2aκ, biotin-FasL (clone MFL3) Armenian hamster IgG2, FITC-streptavidin, and FITC-annexin V were from BD Pharmingen. Monocytes/macrophages were defined as SSC^hi^F4/80^hi^Gr-1lo population, granulocytes as SSC^hi^Gr-1^hi^ cells, and dendritic cells as SSC^lo^F4/80-cells. B cells, T cells, NK cells, and NKT cells were defined as SSC^lo^ CD45R+, SSC^lo^CD3ε+NK1.1-, SSC^lo^CD3ε+NK1.1+, and SSC^lo^CD3ε+NK1.1+, respectively. CD4 T cells (CD4+CD8-), CD8 T cells (CD4-CD8+), and regulatory T cells (CD4+CD8-CD25+) were also counted. Values represent the percentage of the non-parenchymal cells in each organ. 

### 2.12. Statistical Analysis

Data were expressed as the mean ± standard deviation of the means (S.D.). The statistical difference between two variables was calculated by the non-parametric Mann–Whitney U test and between three or more variables by the non-parametric Kruskal–Wallis analysis of variance with Dunn’s test. Statistical analyses were performed using the Graph-pad Prism version 7.0 (Graph-pad Software, San Diego, CA, USA). A *p* < 0.05 was considered statistically significant.

## 3. Results

### 3.1. High Concentration of Ferulic Acid in Culture Supernatant from Bacteria Grown in the Presence of Insoluble Arabinoxylans

We measured the levels of ferulic acid by HPLC-DAD in a control medium containing the simple sugar components of arabinoxylan (arabinose/xylose: Ara/xylose), simple or soluble wheat arabinoxylan (sWAX), or insoluble wheat arabinoxylan (InWAX) and in the culture supernatant from Bacteroides intestinalis (B.i.), cultured on Ara/xylose (i.e., B.i. + Ara/xylose), WAX (i.e., B.i. + sWAX), or InWAX (i.e., B.i. + InWAX). The level of ferulic acid was high in the B.i. + InWAX group, very low in the InWAX group, and it was undetected in the other groups (Figure 1a,b).

### 3.2. The Culture Supernatant from Bacteroides Intestinalis Modulates the Immune Response under Inflammatory Conditions

Under diseased conditions, including diabetes mellitus, cancer, and microbial infection, ferulic acid has been demonstrated to act as a potent anti-inflammatory factor [9]. We, therefore, evaluated the effect of ferulic acid in an inflammatory model prepared in vitro by culturing mouse bone marrow-derived dendritic cells in the presence of LPS. A group of dendritic cells was stimulated with control media containing arabinose/xylose (Ara/xyl), the sugar components of arabinoxylan, simple or soluble wheat arabinoxylan (sWAX), or insoluble wheat arabinoxylan (InWAX) in the presence or absence of LPS. An important difference between sWAX and InWAX is the presence of ferulic acid linkages in InWAX and not in sWAX. Another group of dendritic cells was stimulated with the culture supernatant from *Bacteroides intestinalis* grown in the Ara/xylose medium (Bi + Ara/xylose), *Bacteroides intestinalis* grown in sWAX (B.i. + sWAX), or *Bacteroides intestinalis* grown in InWAX (B.i. + InWAX) in the presence or absence of LPS. As expected, the concentrations of the anti-inflammatory cytokines interleukin-10 (IL-10) and transforming growth factor β1 (TGFβ1) were significantly increased, whereas the concentration of the pro-inflammatory cytokine tumor necrosis factor α (TNFα) was significantly decreased in the cell supernatant from dendritic cells treated with ferulic acid compared to untreated controls (Figure 2). The concentrations of IL-10 and TGFβ1 also significantly increased, while that of TNFα was significantly decreased in the supernatant from dendritic cells cultured in InWAX culture supernatant from *Bacteroides intestinalis* compared to dendritic cells cultured in InWAX medium without *Bacteroides intestinalis* (Figure 2). We attribute the congruent results of the treatment with ferulic acid and the culture supernatant from *Bacteroides intestinalis* grown in InWAX to the release of side-chain ferulic acid during the growth of this colonic bacterium on the complex polysaccharide, as we have previously reported high concentrations of ferulic acid in the culture supernatant from *Bacteroides intestinalis* cultured on insoluble arabinoxylans and also as shown in Figure 1. Thus, the presence of ferulic acid in the bacterial culture supernatant may explain the anti-inflammatory activity of the degraded products of complex arabinoxylans by *Bacteroides intestinalis* [7].

### 3.3. The Culture Supernatant from Bacteroides Intestinalis Modulates the Immune Response under Physiological Conditions

The beneficial effect of dietary arabinoxylans in the healthy population has been well-documented, and this effect has been linked to their antioxidant properties [5,9]. Here, we investigated whether administering ferulic acid can stimulate the immune response under physiological conditions by orally gavaging healthy wild-type mice with the phenolic compound and comparing the effects with that of mice receiving the culture supernatant from *Bacteroides intestinalis* grown on wheat bran or grown on de-starched wheat bran. Each of the two bacterial culture supernatants contained ferulic acid cleaved by the *Bacteroides intestinalis* cells [7]. The administration of ferulic acid significantly increased the relative number of monocytes and granulocytes in both the mesenteric lymph nodes and spleen, an observation recapitulated with the de-starched wheat bran bacterial culture supernatant, which contained high amounts of ferulic acid compared with the wheat bran culture supernatant (Figure 3a,b and Figure 4a,b) [7]. In addition, the oral treatment with ferulic acid increased the percentage of CD8+ T cells and natural killer cells in mesenteric lymph nodes and the percentage of CD4+ T cells in spleen, observations also recapitulated with the destarched wheat bran bacterial culture supernatant (Table 2 and Table 3). The plasma concentrations of the T helper type 1 (Th1) cell-derived cytokines, IL-2 and IL-12 were also significantly increased in mice treated with the ferulic acid or with the bacterial culture supernatant compared to controls (Figure 4c). Importantly, one must note that the destarched wheat bran, with the starch removed, was more similar to the dietary component that flows to the colon, since the starch is digestible by the enzyme amylase secreted in the stomach of the human host. Hence, the de-starched wheat bran likely has more and easily accessible ferulic acid moieties.

### 3.4. The Culture Supernatant from Bacteroides Intestinalis Modulates the Immune State of Colonic Cells

The Bacteroidetes ferment arabinoxylan to short-chain fatty acids and other metabolites, including ferulic acid [7]. To determine how the phenolic compound may impact the host, we used two human cell lines as proxies to examine cytokines’ expression upon treatment with the ferulic acid-containing bacterial culture supernatant. The control experiments showed that ferulic acid increased IL-12, IFNα, and IFNβ in Caco-2 cells in a concentration-dependent manner (Figure 5). Treating the cells with the bacterial culture supernatant, from either wheat bran or destarched wheat bran medium, generally led to increases in the levels of the three cytokines (Figure 5). 

The experiments were repeated with a HIEC-6 cell line, and the effects of ferulic acid on the expression of the three cytokines were similar to those observed with the Caco-2 cells. Treating HIEC-6 cells with the bacterial culture supernatant led to increased IL-12 protein levels (Figure 6). In contrast, significant increases in both transcript and protein levels of IFNβ were observed with the treated cells, irrespective of the source of the bacterial culture supernatant (Figure 6). Increases in the levels of the IFNα transcript were also observed with either bacterial culture supernatant treatment of HIEC-6 cells.

## 4. Discussion

Arabinoxylans are the most abundant hemicellulose polysaccharides of dietary fiber found in fruits, vegetables, and whole grains [5,17]. Humans can digest arabinoxylans only through enzymes secreted by their gut microbiome [18,19]. Several studies have documented that Bacteroidetes, including *Bacteroides intestinalis* and *Bacteroides cellulosilyticus*, are the main degraders of arabinoxylans in the human gut [13,14]. *Bacteroides* spp. have gene clusters called polysaccharide utilization loci that promote the expression of esterases to metabolize arabinoxylans [20]. We recently demonstrated that besides degrading simple or soluble arabinoxylan (i.e., without ferulic acid side chains), *Bacteroides* spp. can degrade complex arabinoxylans, releasing large amounts of ferulic acid side chains [7,17]. The ferulic acid absorption sites are the small and large intestines, but the main site appears to be the human colon, where ferulic acid is released from the food matrix by microbial esterases [21,22].

In vitro and in vivo studies have demonstrated the protective activity of ferulic acid in pathological conditions, including diabetes, allergic inflammation, metabolic syndrome, Alzheimer’s disease, cardiovascular disorders, and cancer [9]. The beneficial effects of ferulic acid have been attributed to its antioxidant function and modulatory activity on the immune system, the inflammatory response, and cell survival [9]. Consistent with the regulatory activity of ferulic acid on host immunity and inflammation under pathological conditions, here we found that both ferulic acid sources (i.e., the raw compound or the form released into bacterial culture supernatant) increased the expression of anti-inflammatory cytokines (TGFβ1, IL-10) and decreased that of TNFα, a pro-inflammatory cytokine, from dendritic cells stimulated with LPS. This observation is consistent with the results reported in previous studies [10]. 

Recent studies have demonstrated that ferulic acid is also involved in maintaining host health status by stimulating the immune system and physiological homeostasis [10]. Supplementation of ferulic acid in the diet increases the number of circulating leukocytes, lysosome activity, serum levels of antioxidant enzymes in fish, enhances leukocytes’ activity in mice, and prevents diabetes mellitus and metabolic syndrome in mice and rats [23,24,25,26,27,28]. Consistent with these previous observations, here we found that administration of the pure compound of ferulic acid and the culture supernatant of the colonic bacterium *Bacteroides intestinalis* grown on complex arabinoxylans enhanced the number of monocytes and granulocytes and the circulating level of the Th1 cytokines IL-12 and IL-2 in healthy mice, further supporting the protective role of ferulic acid under physiological or healthy conditions. However, it is worth noting here that in addition to ferulic acid, other metabolites released from complex arabinoxylan degradation by *Bacteroides intestinalis* may also contribute to the beneficial health effects observed in our model [5].

To gain insights into the cell source of Th1 cytokine expression in the healthy mice, we cultured human intestinal cell lines in the presence of ferulic acid or the culture supernatant from *Bacteroides intestinalis* grown on ferulic acid-rich wheat bran or de-starched wheat bran (a complex arabinoxylan) and evaluated the expression of Th1 cytokines. We found that ferulic acid and the culture supernatant from *Bacteroides intestinalis* increased the expression of IL-12, IFNα, and IFNβ in human intestinal cells. IL-12 belongs to a family of heterodimeric cytokines encoded by the two subunit genes, p35 and p40 genes, that produce a heterodimer (p70) or homodimer (p40) protein [29]. IL-12 is the master inducer of type 1 helper T cells, which play critical roles in protecting against bacterial/viral infection, allergic inflammation, and cancer [30]. The type I interferons IFNβ and IFNα are broadly inducible cytokines involved in the host defense against bacterial/viral infection and cancer immunity [31,32,33,34]. The in vitro and in vivo regulatory activity of the bacterial culture supernatant observed in the present study suggests that ferulic acid released from complex arabinoxylan by *Bacteroides intestinalis* in the gut can protect the host health status by stimulating the immune system.

## 5. Conclusions

This study showed that ferulic acid or the degraded products of complex arabinoxylans by *Bacteroides intestinalis* exerts anti-inflammatory activity in dendritic cells under inflammatory conditions and enhances the Th1 type immune response under physiological conditions in mice.

## Figures and Tables

**Figure 1 microorganisms-09-01126-f001:**
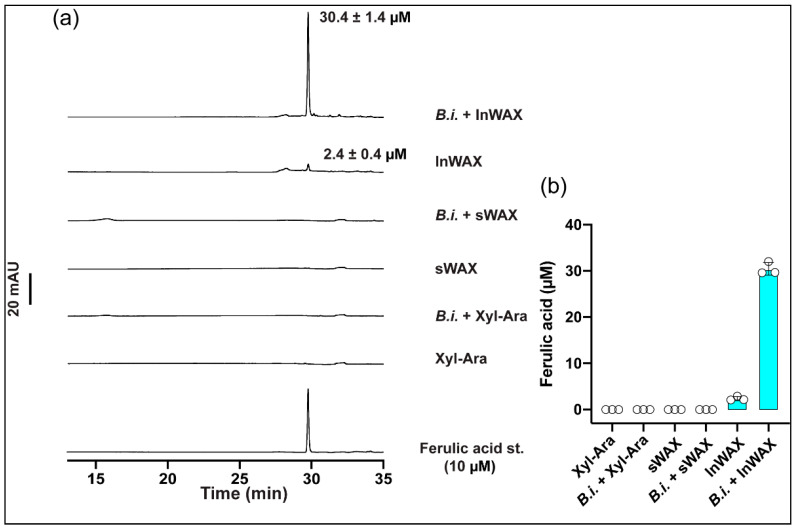
Growth of *Bacteroides intestinalis* in minimal medium containing various substrates as the sole carbon source. (**a**) HPLC-DAD chromatogram of *Bacteroides intestinalis* culture supernatant from 24 h growth on monosaccharides (xylose + arabinose), soluble wheat arabinoxylan (sWAX), and insoluble wheat arabinoxylan (InWAX) as the sole carbon sourcesm showing the release of ferulic acid into the medium only on InWAX. (**b**) Quantification of ferulic acid released into the culture supernatant after 24 h growth. The results are mean ± S.D. of three replicates.

**Figure 2 microorganisms-09-01126-f002:**
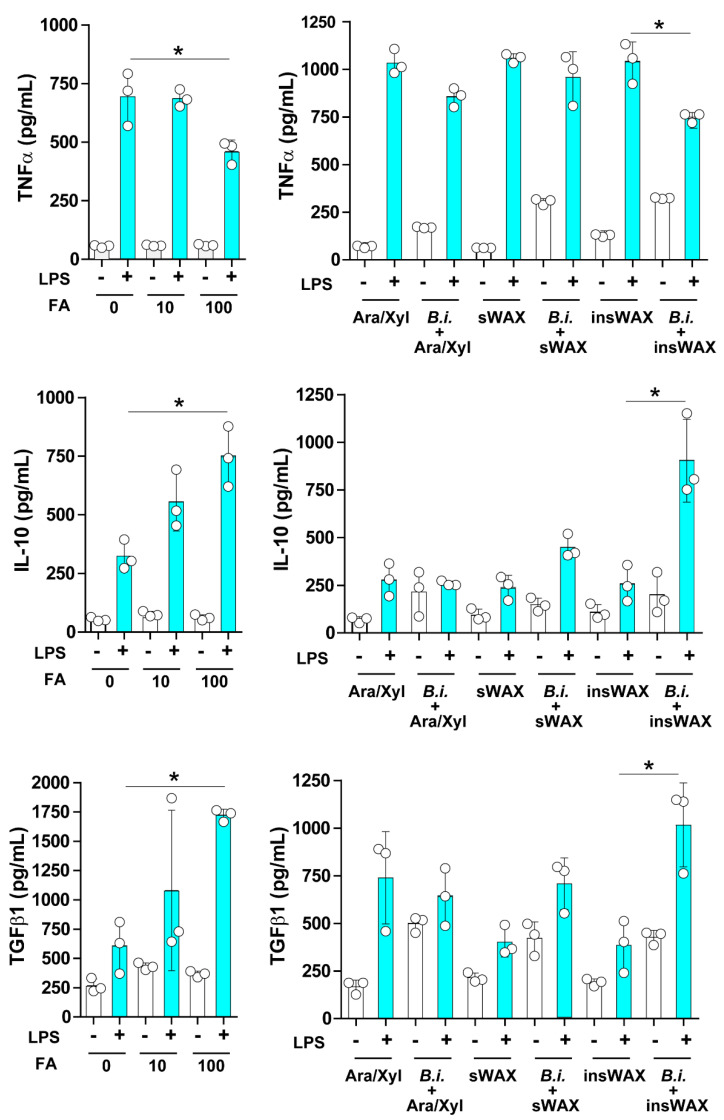
The culture supernatant from *Bacteroides intestinalis* modulated the inflammatory response under inflammatory conditions. Bone marrow-derived cells were isolated from healthy C57BL/6 mice, cultured in vitro, and differentiated to dendritic cells in the presence of Flt3. Dendritic cells were then cultured in a medium containing several concentrations (0, 10, 100 µM) of pure ferulic acid or arabinose/xylose (Ara/xylose), soluble wheat arabinoxylan (sWAX) or insoluble WAX (InWAX), or cultured in the presence of a culture supernatant from *Bacteroides intestinalis* grown with Ara/xylose (*B.i.* + Ara/xylose), sWAX (*B.i.* + sWAX), or InWAX (*B.i.* + InWAX). Maturation of dendritic cells was induced with a lipopolysaccharide (LPS; 100 ng/mL). Tumor necrosis factor α (TNFα), interleukin-10 (IL-10), and transforming growth factor β1 (TGFβ1) were measured by enzyme immunoassays. Data are the mean ± S.D. Results from one representative experiment of two independent experiments are shown. Statistical analysis was performed using the Kruskal–Wallis analysis of variance with Dunn’s test and the Mann–Whitney U test. * *p* < 0.05.

**Figure 3 microorganisms-09-01126-f003:**
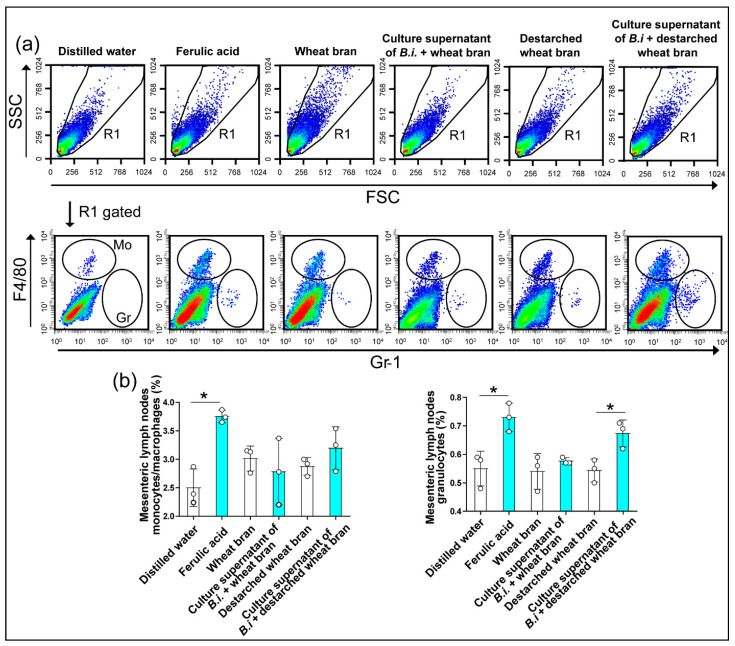
The culture supernatant from *Bacteroides intestinalis* increased monocytes/macrophages and granulocytes in the mesenteric lymph nodes under physiological conditions. Flow cytometry analyses demonstrated an increased percentage of monocytes/macrophages and granulocytes in mesenteric lymph nodes (**a**,**b**) from healthy mice treated with ferulic acid or a culture supernatant of *Bacteroides intestinalis* (B.i.) + destarched wheat bran. The results are shown as the mean ± S.D. of one experiment. Three mice in each treatment group. Statistical analysis was performed using the Mann–Whitney U test. * *p* < 0.05. SSC, side scatter; FSC, forward scatter.

**Figure 4 microorganisms-09-01126-f004:**
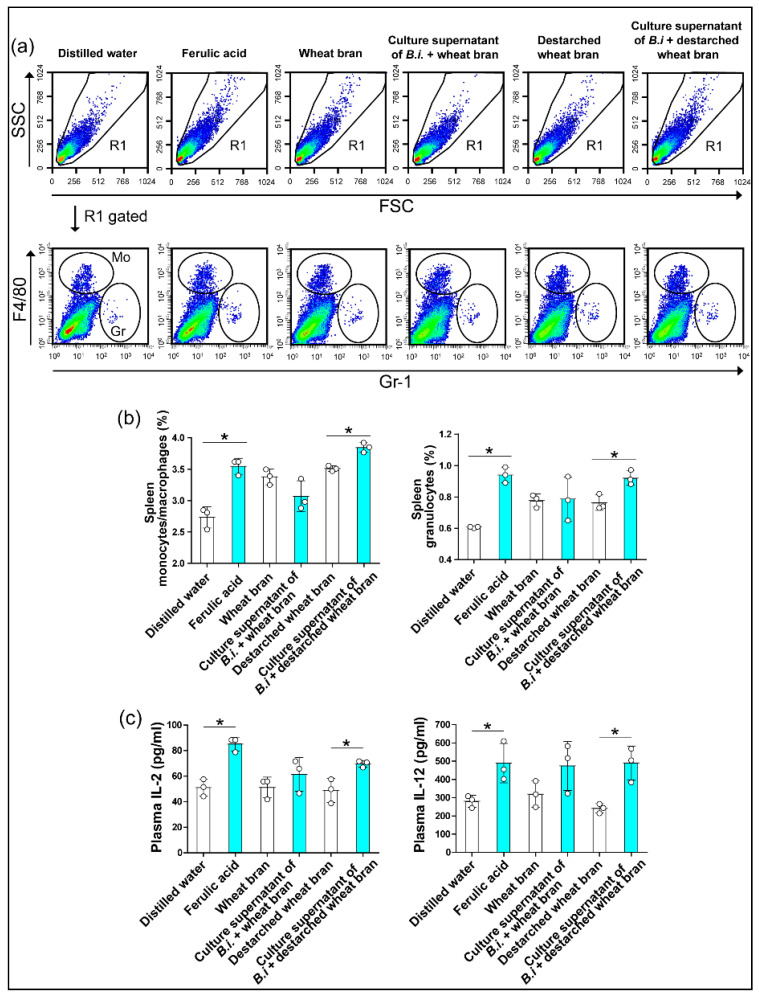
The culture supernatant from *Bacteroides intestinalis* increased monocytes/macrophages and granulocytes in the spleen under physiological conditions. Flow cytometry analyses demonstrated an increased percentage of monocytes/macrophages and granulocytes in the spleen (**a**,**b**) from healthy mice treated with ferulic acid or a culture supernatant of *Bacteroides intestinalis* (B.i) + destarched wheat bran. Enzyme immunoassays showed increased circulating levels of IL-2 and IL-12 in healthy mice treated with ferulic acid and culture supernatant from *Bacteroides intestinalis* grown on destarched wheat bran (**c**). The results are shown as the mean ± S.D. of one experiment. Three mice in each treatment group. Each dot represents one mouse. Statistical analysis was performed using the Mann–Whitney U test. * *p* < 0.05. SSC, side scatter; FSC, forward scatter.

**Figure 5 microorganisms-09-01126-f005:**
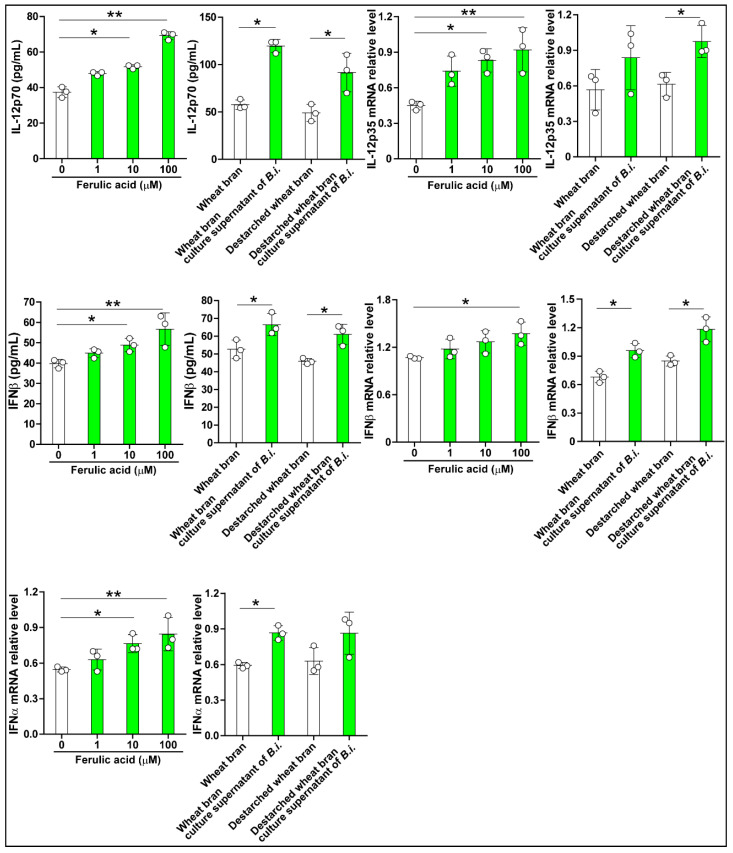
*Bacteroides intestinalis* culture medium modulated cytokine expression in Caco-2 cells. Culture medium from *Bacteroides intestinalis* grown in wheat bran and destarched wheat bran increased the production of cytokines as demonstrated by RT-PCR and enzyme immunoassays. Ferulic acid in three concentrations (1, 10, and 100 μM) was used as a positive control. N = 3 in each treatment group. Data are the mean ± S.D. Results from one representative experiment of two independent experiments are shown. Statistical analysis was performed using the Kruskal–Wallis analysis of variance with Dunn’s test and the Mann–Whitney U test. * *p* < 0.05, ** *p* < 0.01.

**Figure 6 microorganisms-09-01126-f006:**
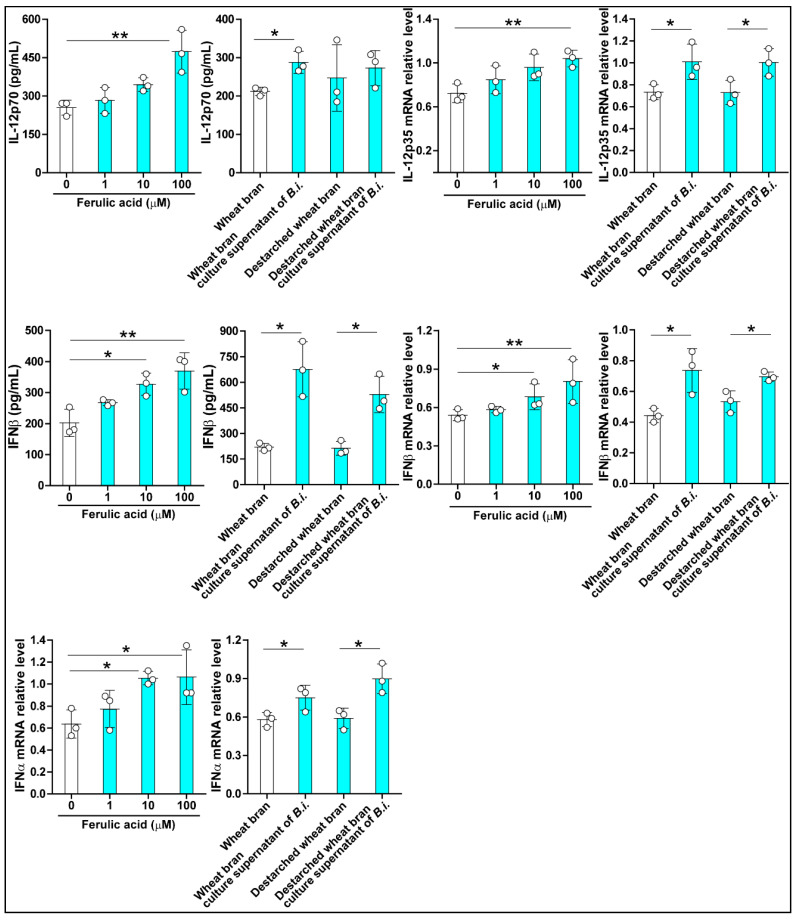
*Bacteroides intestinalis* culture supernatant modulated cytokine expression in HIEC-6 cells. Culture supernatant from *Bacteroides intestinalis* grown in wheat bran and de-starched wheat bran increased the expression of cytokines as demonstrated by RT-PCR and immunoassays. Ferulic acid in three different concentrations (1, 10, and 100 μM) was used as a positive control. N = 3 in each treatment group. Data are the mean ± S.D. Results from one representative experiment of two independent experiments are shown. Statistical analysis was performed using the Kruskal–Wallis analysis of variance with Dunn’s test and the Mann–Whitney U test. * *p* < 0.05, ** *p* < 0.01.

**Table 1 microorganisms-09-01126-t001:** PCR primer sequences for cytokines expression analysis.

Gene	Direction	Sequence (5’ to 3’)	Length	Product Size	Cycles
GAPDH	Sense Antisense	GGAGCGAGATCCCTCCAAAAT GGCTGTTGTCATACTTCTCATGG	21 23	197 bp	26
IL-12 p35	Sense Antisense	CCTTGCACTTCTGAAGAGATTGA ACAGGGCCATCATAAAAGAGGT	23 22	181 bp	35
IFNα	Sense Antisense	GACTCCATCYTGGCTGTGA TGATTTCTGCTCTGACAACCT	19 21	103 bp	33
IFNβ	Sense Antisense	GCTTGGATTCCTACAAAGAAGCA ATAGATGGTCAATGCGGCGTC	23 21	166 bp	33

IFNβ, interferonβ; IFNα, interferonα; IL-12, interleukin-12; GAPDH, glyceraldehyde 3-phosphate dehydrogenase.

**Table 2 microorganisms-09-01126-t002:** Cells in mouse mesenteric lymph nodes.

Treatment	Dendritic Cells (%)	Total Lymphocytes (%)	B Cells (%)	CD4+ T Cells (%)	CD8+ T Cells (%)	Natural Killer Cells (%)	Natural Killer T Cells (%)
Distilled water	2.44 ± 0.33	77.36 ± 3.59	56.63 ± 0.75	11.30 ± 2.69	7.75 ± 1.37	0.29 ± 0.05	0.09 ± 0.02
Ferulic acid	3.23 ± 0.19 *	86.06 ± 1.34 *	53.03 ± 0.98 *	18.87 ± 0.46 *	11.25 ± 0.38 *	0.48 ± 0.03 *	0.13 ± 0.02 *
Wheat bran	2.39 ± 0.11	82.93 ± 0.73	55.62 ± 0.36	15.49 ± 0.86	10.23 ± 0.29	0.31 ± 0.05	0.13 ± 0.01
Culture supernatant of *B.i* + wheat bran	2.97 ± 0.21 †	84.31 ± 0.70	50.86 ± 0.93 †	16.33 ± 0.58	15.19 ± 1.22 †	0.52 ± 0.04 †	0.11 ± 0.02
Destarched bran	3.08 ± 0.18	81.40 ± 1.26	52.71 ± 0.63	14.17 ± 1.13	12.79 ± 0.60	0.46 ± 0.06	0.13 ± 0.03
Culture supernatant of *B.i* + destarched wheat bran	3.14 ± 0.24	82.25 ± 1.38	49.88 ± 0.82 ‡	16.24 ± 1.86	14.01 ± 0.35 ‡	0.64 ± 0.07 ‡	0.12 ± 0.01

Values represent the percentage of total cells and are expressed as the mean + standard errors of the mean. N = 3 mice. Statistical analysis by Mann–Whitney U test. * *p* < 0.05 vs. distilled water; † *p* < 0.05 vs. wheat bran; ‡ *p* < 0.05 vs. destarched bran. *B.i*, *Bacteroides intestinalis.*

**Table 3 microorganisms-09-01126-t003:** Spleen cells in each group of mice.

Treatment	Dendritic Cells (%)	Total Lymphocytes (%)	B Cells (%)	CD4+ T Cells (%)	CD8+ T Cells (%)	Natural Killer Cells (%)	Natural Killer T Cells (%)
Distilled water	3.08 ± 0.06	61.84 ± 2.58	37.12 ± 1.51	11.95 ± 1.02	10.90 ± 0.45	0.42 ± 0.04	0.15 ± 0.03
Ferulic acid	3.94 ± 0.14 *	61.19 ± 1.42	33.32 ± 0.66 *	13.77 ± 0.79 *	12.01 ± 0.21 *	0.61 ± 0.09 *	0.18 ± 0.05 *
Wheat bran	3.19 ± 0.17	57.05 ± 1.88	33.24 ± 0.70	11.32 ± 0.83	10.55 ± 0.42	0.65 ± 0.02	0.12 ± 0.04
Culture supernatant of *B.i* + wheat bran	3.19 ± 0.27	57.41 ± 0.48	31.67 ± 0.45 †	12.57 ± 0.11 †	11.08 ± 0.62	0.66 ± 0.01	0.12 ± 0.02
Destarched bran	3.40 ± 0.16	56.47 ± 0.35	30.10 ± 0.55	13.72 ± 0.66	10.77 ± 0.09	0.61 ± 0.08	0.12 ± 0.01
Culture supernatant of *B.i* + destarched wheat bran	3.36 ± 0.12	56.66 ± 1.64	30.46 ± 1.00	12.52 ± 0.17 ‡	11.48 ± 0.74	0.71 ± 0.06	0.15 ± 0.01

Values represent the percentage of total splenocytes and are expressed as the mean + standard errors of the mean. N = 3 mice. Statistical analysis by Mann–Whitney U test. * *p* < 0.05 vs. distilled water; † *p* < 0.05 vs. wheat bran; ‡ *p* < 0.05 vs. destarched bran. *B.i*, *Bacteroides intestinalis.*

## Data Availability

All data are available from the corresponding author upon reasonable request. After publication the raw data will be available at the Office of the Mie University Graduate School of Medicine.

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
