# Peer review of "Degradation Products of Complex Arabinoxylans by *Bacteroides intestinalis* Enhance the Host Immune Response"

_microorganisms, 2021, doi:10.3390/microorganisms9061126_

Round 1
Reviewer 1 Report
The authors show that ferulic acid or the degraded products of complex arabinoxylans by B. intestinalis exerts anti-inflammatory activity in dendritic cells and enhances the Th1 type immune response in mice. The authors need to be clearer on the experiments they performed. Do they show the mean of three independent experiments? It is difficult for me to conclude as it is not clear how reproducible are the results.
They should rather use non-parametric tests (ie Mann-Whitney instead of T-test student) as they are only few dots. Also be careful to add a title to all X axes, and to FACS plots.
L68 Bacteroides intestinalis in italics – also everywhere in the text
L195 The term “culture supernatant” is preferable to “spent medium”
L240 Figure 2 To what correspond the three dots? Is it one experiment representative of three? Or each dot is the mean obtained from one experiment?
L251 From Fig3, why do you use “de-starched wheat bran” instead of InsWAX used in Figures 1 and 2? Please explain.
L271 Figure 3 Axes of cytometry plots are missing. Legends of X axes are missing on the graphs. Please remove the title to give information on the X axis
L277 Table 2 and 3 are not clear
L313 Figure 4 No legend for X axes on graphs. “Data are the mean” what do you mean? Mean of three independent experiments or one representative experiment out of three? How mice did you use? Only three? This is not clear.
L366 In the conclusion, the authors use the terms “under pathological condition” to refer to their condition of LPS stimulation. I would rather use “under inflammatory conditions”.
The authors should test if ferulic acid has in impact on disease development, on a model of their choice, to confirm that this molecule has a beneficial effect in mice. It would increase the impact of your finding if you can show that the immunomodulatory activity protect from a pathological condition.
Author Response
Response to Reviewer 1
Comment 1
The authors show that ferulic acid or the degraded products of complex arabinoxylans by B. intestinalis exerts anti-inflammatory activity in dendritic cells and enhances the Th1 type immune response in mice. The authors need to be clearer on the experiments they performed. Do they show the mean of three independent experiments? It is difficult for me to conclude as it is not clear how reproducible are the results.
Response
We very much appreciate the comments of the Reviewer that substantially improved the quality of the manuscript. We have clarified in the legends the number of experiments.
Please see page 8, line 305, and lines 331 to 332; page 9, line 356; page 12, lines 438 to 439 in the revised manuscript.
Comment 2
They should rather use non-parametric tests (ie Mann-Whitney instead of T-test student) as they are only few dots.
Response
As suggested by the Reviewer, we re-analyzed the data using non-parametric statistically methods: the Mann-Whitney U test for evaluating the difference between two variables, and the Kruskal-Wallis analysis of variance with Dunn’s test to compare three or more variables.
Please see page 5, lines 220 to 222 under Statistical analysis, and the legends of Figure 2 (Lines 305 to 306), Figure 3 (lines 332 to 333), Figure 4 (lines 357 to 358), Tables 2 (line 375 to 384) and 3 (385 to 395) on page 10, Figure 5 (lines 439 to 440), Figure 6 (lines 466 to 467).
Comment 3
Also be careful to add a title to all X axes, and to FACS plots.
Response
We have added all X and Y axes for the FACS plots and splitted the Figure 3 of the old version of the manuscript into Figure 3 and Figure 4 to make them more comprehensible and to show the gating strategy used.
Please see Figure 3 on page 8 and Figure 4 on page 9.
Comment 4
L68 Bacteroides intestinalis in italics – also everywhere in the text
L195 The term “culture supernatant” is preferable to “spent medium”
L240 Figure 2 To what correspond the three dots? Is it one experiment representative of three? Or each dot is the mean obtained from one experiment?
Response
We have changed “Bacteroides intestinalis” to italic form, and used “culture supernatant” instead of “spent medium” through all the text in the revised manuscript.
For Figure 2, a preliminary experiment and then a second experiment was performed. We have shown the representative results of the two experiments.
Please see the figure legend of Figure 2 on page 8, line 305 in the revised manuscript.
Comment 5
L251 From Fig3, why do you use “de-starched wheat bran” instead of InsWAX used in Figures 1 and 2? Please explain.
Response
There is an explanation for the reason we used “de-starched wheat bran” instead of InsWAX in Figures 3, 4, 5 and 6.
Please see page 10, lines 374 and lines 396 to 398.
Also, we want to apologize for not mentioning the method we used to prepared the de-starched wheat bran in the previous version of the manuscript. We have added a new section (2.4) under Methods describing how we prepared the de-starched wheat bran.
Please see that section (Arabinoxylan substrates) on page 3, lines 97 to 107 in the revised manuscript.
Comment 6
L271 Figure 3 Axes of cytometry plots are missing. Legends of X axes are missing on the graphs. Please remove the title to give information on the X axis
Response
We have splitted the old Figure 3 into Figure 3 and Figure 4 to make them clearer. We added the X axes, and the legends of the X and Y axes.
We have also shown the gating strategy used.
Please see Figures 3 and 4 on pages 8 and 9, respectively, in the revised manuscript.
Comment 7
L277 Table 2 and 3 are not clear
Response
We have modified the two tables to make them clearer.
Please see Table 2 and Table 3 on page 10, and their descriptions on pages 9 and 10 lines 367 to 371 in the revised manuscript.
Comment 8
L313 Figure 4 No legend for X axes on graphs. “Data are the mean” what do you mean? Mean of three independent experiments or one representative experiment out of three? How mice did you use? Only three? This is not clear.
Response
The experiment described in the old Figure 4 (Figure 5 in the revised manuscript) has been done using the intestinal Caco-2 cell lines. The figure shows the representative results of one experiment from two independent experiments. The same explanation applied to the old Figure 5 (Figure 6 in the revised manuscript).
Please see the legends of the Figures 5 (pages 12) and Figure 6 (page 12) of the revised manuscript.
Comment 9
L366 In the conclusion, the authors use the terms “under pathological condition” to refer to their condition of LPS stimulation. I would rather use “under inflammatory conditions”.
Response
A suggested by the Reviewer, we replaced the expression “under pathological condition” by “under inflammatory conditions”
Comment 10
The authors should test if ferulic acid has in impact on disease development, on a model of their choice, to confirm that this molecule has a beneficial effect in mice. It would increase the impact of your finding if you can show that the immunomodulatory activity protect from a pathological condition.
Response
In the present study we used an in vitro model (dendritic cells) to demonstrate the role of ferulic acid under inflammatory conditions because many previous studies have already demonstrated the beneficial effects of ferulic acid in experimental animal models of the disease.

Reviewer 2 Report
The manuscript by Yasuma et al. reports on both in vitro and in vivo studies in mice that demonstrates that ferulic acid and the Bacteroides intestinalis degradation products of complex arabinoxylans exert anti-inflammatory activity in dendritic cells challenged with LPS. Furthermore cell-line studies in both Caco-2 cells and HIEC-6 cells demonstrated effects on cytokine expression linked to ferulic acid. Overall, the manuscript is very well written and easy to follow and I have only very few specific comments and suggestions as outlined below.
L60 Should read: "ferulic acid"
L69 "Bacteroides intestinalis" should be italics here and throughout the manuscript
Figure 1 Please make sure abbreviations are consistent (eg. sWAX)
Figure 2 Legend should include details of the left panels as well (adding pure ferulic acid)
L345 It could be further discussed that other metabolites of Bacteroides may also be important.
Author Response
Response to Reviewer 2
Comment 1
The manuscript by Yasuma et al. reports on both in vitro and in vivo studies in mice that demonstrates that ferulic acid and the Bacteroides intestinalis degradation products of complex arabinoxylans exert anti-inflammatory activity in dendritic cells challenged with LPS. Furthermore cell-line studies in both Caco-2 cells and HIEC-6 cells demonstrated effects on cytokine expression linked to ferulic acid. Overall, the manuscript is very well written and easy to follow and I have only very few specific comments and suggestions as outlined below.
Response
We very much appreciate the comments of the Reviewer that significantly improved the quality of the manuscript.
Comment 2
L60 Should read: "ferulic acid"
L69 "Bacteroides intestinalis" should be italics here and throughout the manuscript
Response
We have corrected the mistakes on the expression of ferulic acid.
Please see page 2, line 67 in the rvised manuscript.
We have also changed to italic "Bacteroides intestinalis" throughout the manuscript as suggested by the Reviewer.
Comment 3
Figure 1 Please make sure abbreviations are consistent (eg. sWAX)
Response
We have made consistent the abbreviations in Figure 1.
Please see Figure 1 on page 6 in the revised manuscript.
Comment 4
Figure 2 Legend should include details of the left panels as well (adding pure ferulic acid)
Response
We have added the explanation on ferulic acid as suggested by the Reviewer.
Please see the legend of Figure 2 on page 8, lines 300 to 301.
Comment 5
L345 It could be further discussed that other metabolites of Bacteroides may also be important.
Response
We have added a sentence to the discussion section and cited a new reference in the revised manuscript.
Please see page 13, lines 502 to 505.

Round 2
Reviewer 1 Report
The authors made appropriate modifications.
Please l305, 439 and 466, add two "independant" experiments.
For the other figures, only one experiment has been performed. I remind that a minimum of three independant in vitro experiments are usually required.
N=3 mice for in vivo experiments is really low.
I leave the final decision to the editor.